# Parental Sleep, Distress, and Quality of Life in Childhood Acute Lymphoblastic Leukemia: A Longitudinal Report from Diagnosis up to Three Years Later

**DOI:** 10.3390/cancers14112779

**Published:** 2022-06-03

**Authors:** Niki Rensen, Lindsay Steur, Martha Grootenhuis, Jos Twisk, Natasha van Eijkelenburg, Inge van der Sluis, Natasja Dors, Cor van den Bos, Wim Tissing, Gertjan Kaspers, Raphaële van Litsenburg

**Affiliations:** 1Princess Máxima Center for Pediatric Oncology, 3584 CS Utrecht, The Netherlands; n.rensen@prinsesmaximacentrum.nl (N.R.); m.a.grootenhuis@prinsesmaximacentrum.nl (M.G.); n.k.a.vaneijkelenburg@prinsesmaximacentrum.nl (N.v.E.); i.m.vandersluis@prinsesmaximacentrum.nl (I.v.d.S.); n.dors@prinsesmaximacentrum.nl (N.D.); c.vandenbos-5@prinsesmaximacentrum.nl (C.v.d.B.); w.j.e.tissing@prinsesmaximacentrum.nl (W.T.); g.j.l.kaspers@prinsesmaximacentrum.nl (G.K.); 2Emma Children’s Hospital, Amsterdam UMC, VU University, 1081 HV Amsterdam, The Netherlands; l.steur@amsterdamumc.nl; 3Department of Epidemiology and Biostatistics, Amsterdam UMC, 1081 HV Amsterdam, The Netherlands; jwr.twisk@amsterdamumc.nl; 4Sophia Children’s Hospital, Erasmus Medical Center, 3015 CN Rotterdam, The Netherlands; 5Amalia Children’s Hospital, Radboud University Medical Center, 6525 GA Nijmegen, The Netherlands; 6Emma Children’s Hospital, Amsterdam UMC, University of Amsterdam, 1105 AZ Amsterdam, The Netherlands; 7Beatrix Children’s Hospital, University Medical Center Groningen, 9713 GZ Groningen, The Netherlands

**Keywords:** psychosocial, quality of life, ALL, parents, sleep

## Abstract

**Simple Summary:**

This study assessed sleep problems, distress, and quality of life in parents of children with the most common form of childhood cancer, acute lymphoblastic leukemia (ALL). Parents completed questionnaires at different time points, shortly after diagnosis until 1 year after end of treatment. Before this study, longitudinal research on parental psychosocial functioning, especially sleep problems, was sparse. This study shows that although parental functioning improves over time, 33% of parents still report sleep problems three years after their child’s diagnosis. Half of those parents also report clinical distress. Presence of sleep problems and distress negatively affects quality of life over time. Vulnerable parents are those who experience little social support or parenting problems, report a chronic illness for themselves or pain for their child, have a child with higher risk ALL, and are closer to diagnosis. This study stresses the importance of systematically monitoring parental functioning both throughout and after treatment—including sleep.

**Abstract:**

This study assessed sleep, distress and quality of life (QoL) in parents of children with acute lymphoblastic leukemia (ALL) from diagnosis to three years after, and the impact of sleep and distress on QoL. Additionally, this study explored determinants of sleep and distress. Parents completed the MOS Sleep, Distress Thermometer for Parents and SF-12 at four-five months (T0), one year (T1), two years (T2), and three years (T3) after diagnosis. The course of outcomes and longitudinal impact of clinically relevant sleep problems (>1SD above reference’s mean) and clinical distress (score ≥ 4) on QoL Z-scores were assessed with linear mixed-models. Determinants of sleep and distress were assessed with multinomial mixed-models. Parents (81% mothers) of 139 patients (60% males; 76% medium-risk (MR)) participated. Distress and QoL gradually restored from T0 to T3. Sleep problems improved, but were still elevated at T3: 33% reported clinically relevant sleep problems, of which 48% in concurrence with distress. Over time, presence of sleep problems or distress led to lower mental QoL Z-scores (SD-score −0.2 and −0.5, respectively). Presence of both led to a cumulatively lower Z-score (SD-score −1.3). Parents in the latter group were more likely to report insufficient social support, parenting problems, a chronic illness, pain for their child, having a child with MR-ALL, and being closer to diagnosis. In conclusion, parental well-being improves over time, yet sleep problems persist. In combination with ongoing distress, they cumulatively affect QoL. Special attention should be given to parents who are vulnerable to worse outcomes.

## 1. Introduction

Acute lymphoblastic leukemia (ALL) is the most common type of childhood cancer. Pediatric cancer diagnoses significantly impact the psychosocial well-being of children and their families [1,2]. Parents are at risk for quality of life (QoL) impairment [3,4]. Major determinants of adverse QoL outcomes in this group are sleep problems [3] and ongoing psychological distress [5,6]. Impaired sleep seems to be a common issue in parents during treatment for ALL [7,8,9]. In fact, in a large cohort of parents after completion of their child’s cancer treatment, the proportion of sleep problems was still 37% as well (compared to 16% in the general population) [10].

We have previously proposed a conceptual, biopsychosocial model of insomnia complaints in parents of children with cancer [11], based on the model of Spielman [12]. This model distinguishes predisposing, precipitating, and perpetuating factors. Predisposing factors are for example a chronic illness, low socioeconomic status, and prior episodes of sleep problems [13]. Precipitating factors are almost always stressful events [14], in this case the child’s cancer diagnosis and following novel treatment-related stressors. Sleep problems are often left untreated and thus tend to become chronic [15]. Perpetuating factors may then be dysfunctional sleep habits that parents develop over the years, and circadian rhythm disturbances [12,15]. Furthermore, chronic distress may be a perpetuating factor.

Findings from the general population affirm that sleep and distress are closely related [16,17]. Additionally, a previous pediatric oncology study showed that the majority of parents with sleep problems experienced (ongoing) distress as well [10]. In general, within pediatric oncology, parental distress peaks during the first months and then gradually declines. It is known, however, that a proportion of parents experience ongoing distress for years [6].

Sleep and distress are potentially modifiable and therefore a target to improve parental and child well-being. However, the longitudinal course of parental sleep problems in pediatric oncology has never been investigated, and the specific contribution of persisting distress is unclear. Similarly, risk factors for these symptoms and their impact on QoL over time are not well-known.

Therefore, the main aims of this study are:(1)To assess the longitudinal course of sleep problems, distress and QoL in parents of children with ALL, from diagnosis up to three years later.(2)To assess the influence of experiencing sleep problems and/or clinical distress levels on parental QoL over time.(3)To longitudinally explore determinants (sociodemographic, medical, and psychosocial) of experiencing sleep problems and/or distress over time.

## 2. Materials and Methods

### 2.1. Study Population

This study included participants from the SLAAP [SLEEP]-study (SLeep in children with Acute lymphoblastic leukemia And their Parents), as described in detail elsewhere [18]. Families were eligible if the child was at least 2 years old, diagnosed with ALL for the first time, and received treatment according to the ALL11 protocol in the Netherlands. Furthermore, families had to have sufficient knowledge of the Dutch language to complete questionnaires independently. Eligibility was assessed by the treating physician.

### 2.2. Overview of ALL11 Protocol and Study Measurements

Figure 1 gives a schematic overview of the ALL11 treatment protocol [19] and timing of the SLAAP-study measurements, specified by risk group.

The ALL11 treatment protocol distinguishes three risk groups: standard risk (SR, ±25% of patients), medium risk (MR, ±70%), and high risk (HR, ±5%). The first phase of treatment (i.e., induction and consolidation) is generally the same across risk groups. After intensification, SR and MR-patients progress into maintenance treatment, whereas HR-patients follow a different treatment regimen (intensive chemotherapy or an allogenic stem cell transplant). MR maintenance is more intensive than SR maintenance, with weekly hospital-administered intravenous chemotherapy versus only oral chemotherapy at home, respectively. Furthermore, patients in MR maintenance receive high dose of dexamethasone pulses (6 mg/m^2^/day) during most of their treatment. Treatment duration for most patients with SR or MR ALL is two years, but three years for MR patients with an IKZF1 deletion (±12% of patients in the MR group).

Parents completed questionnaires on their sleep, distress, and QoL at four time points. All measurements took place in the home setting and included the same study elements. The first measurement (T0) took place after ALL induction therapy, scheduled between high-dose methotrexate courses for which hospitalization was required. Patients did not receive glucocorticoids in this time period. The second measurement (T1) was planned during maintenance treatment, approximately one year after diagnosis. Parents of MR-patients completed the questionnaires twice at this time point: once in a week that their child received dexamethasone, and once in a week without. For the longitudinal results that are described here, only the off-dexamethasone measurements were taken into account, because of the potential adverse effects of dexamethasone on parental functioning. Parents of SR and HR-patients completed one measurement at T1. The third measurement (T2) was scheduled approximately two years after diagnosis, around end of treatment for the majority of patients (except for patients with an IKZF1 deletion and a third year of treatment (±12% of patients in the MR group)). Finally, the last measurement (T3) took place approximately three years after diagnosis, one year after end of treatment completion for most patients.

### 2.3. Questionnaires

#### 2.3.1. Sociodemographic Characteristics

Parents completed a general questionnaire to assess their age, sex, and education (the highest educational level of both parents was taken into account, [20] to give an indication of families’ socioeconomic status). Furthermore, parents provided information on child’s age, sex, pain in the last week (rated from 0 (no pain) to 10 (most severe pain); parent-reported), and child’s comorbidities (parent-reported). The DCOG provided information on date of diagnosis, risk group stratification, and presence of IKZF1 gene deletion (patients with this mutation receive a third year of treatment).

#### 2.3.2. Sleep

Sleep was assessed with the Medical Outcomes Study (MOS) Sleep Scale, which is a one-week retrospective questionnaire [21]. The 9-item sleep problem index (SLP-9) was constructed according to the MOS manual and included for analyses [22]. This sum score includes 9 of the 12 items of the MOS, amongst which all items on sleep disturbance, sleep adequacy, and daytime somnolence—thus representing insomnia symptoms. Dutch reference values were used for comparison [23].

#### 2.3.3. Distress and Psychosocial Factors

Distress was measured with the Distress Thermometer for Parents (DT-P) [24]. This instrument consists of a “thermometer” on which parents score their overall distress in the past week (range 0–10, with 4 being the cut-off for clinical distress), five separate items, and items on several problem domains (practical, social, emotional, physical, cognitive, parenting). The thermometer score and parenting problem subscale (dichotomized as 0 versus at least 1 problem) were included for analyses, as well as the separate items on social support, chronic illness, and wish for referral (dichotomized as yes/maybe versus no). Dutch reference values were available [25].

#### 2.3.4. Quality of Life

The Short Form-12 (SF-12) was employed to evaluate QoL. This brief, generic QoL questionnaire measures physical and mental well-being (one-week recall period) by using norm-based scoring [26]. The physical component summary score (PCS) and mental component summary score (MCS) were included and Dutch reference values were available [27].

### 2.4. Statistical Analysis

#### 2.4.1. Study Population

Baseline characteristics were described. For time-dependent variables, characteristics were provided per time point.

#### 2.4.2. Longitudinal Course of Sleep, Distress, and QoL

The longitudinal courses of sleep, distress, and QoL were assessed with linear mixed-model analysis, with random intercept on child’s level. To specifically assess change between the different measurements, time points were added as categorical covariate. Additionally, analyses were corrected for parent’s sex. The latter was done because unfortunately, the parent respondent differed at least one time across the measurements in about 20% of families. To overcome this, a sensitivity analysis was performed with inclusion of only the measurements that were completed by the same parent (which excluded seven cases and 26 single measurements). Yet, these analyses revealed very similar results to when all measurements of all cases were included with a correction for parent’s sex. Therefore, we chose to include all measurements instead of disposing data that parents provided. Effects were shown as regression coefficients with 95% confidence interval (C.I.)

#### 2.4.3. Predictive Determinants of Sleep and Distress

In accordance with our previous work, four categories of parents were distinguished, based on the presence or absence of sleep problems and/or distress [10]. Clinically relevant sleep problems were defined as an SLP-9 score >1SD above the Dutch population’s mean and clinical distress as a score ≥4 [24].

To assess determinants of having sleep problems and/or distress over time, multinomial mixed-models were built (random intercept on child level). The reference category consisted of the group of parents with neither sleep problems nor distress. Parent’s sex was included in the model by default. The following variables were included as potential predictors: presence of parental chronic illness, family’s highest educational level (dichotomized as low or middle vs. high), experienced social support, parenting problems, child’s sex, risk group stratification (dichotomized as standard risk vs medium or high risk), time since diagnosis, parent-rated child’s pain (dichotomized as clinical (≥4) or non-clinical (<4) [28]), and child’s age at time of measurement. Backward selection with preselection (*p*-value < 0.15 in univariate analysis) was used to build the final multivariable model. In the final model, a *p*-value of <0.10 was considered to be significant. Effects were shown as odds ratio (OR) with 95% C.I.

#### 2.4.4. Relationships between Sleep, Distress, and QoL

Finally, the influence of being in each one of the sleep/distress categories on MCS and PCS z-scores over time was assessed with linear mixed-model analysis, with random intercept on child’s level. The four categories were added as categorical covariate and analyses were corrected for parent’s sex.

All analyses were done with IBM SPSS Statistics version 26.0.

## 3. Results

### 3.1. Study Population

One hundred fifty-one families provided written informed consent (response rate 67%), of which 139 completed questionnaires at one or more time points. Table 1 summarizes patients’ baseline characteristics. Table 2 displays parents’ characteristics and their unadjusted sleep, distress, and QoL scores per time point.

### 3.2. Longitudinal Course of Sleep, Distress and QoL

Figure 2 shows the longitudinal course of SLP-9, distress, MCS and PCS scores across time points, and the corresponding Dutch reference values. The SLP-9 score (sleep) decreased from 38.3 [35.2–41.4] at T0 to 31.1 [27.8–31.4] at T3 (*p* < 0.001), with a significant improvement between T1 and T2 as well (mean difference −3.3 [−6.4; −0.2], *p* = 0.036). Distress scores gradually improved between all time points, from 5.6 [5.1; 6.1] at T0, to 4.7 [4.2; 5.2] at T1, to 4.0 [3.4; 4.5] at T2, and finally 2.5 [1.9; 3.1] at T3. MCS Z-scores (mental QoL) also gradually improved from T0 (−0.9 [−1.1; −0.7]) to T3 (0.12 [−0.1; 0.3]). PCS scores (physical QoL) were never impaired and did not change over time.

### 3.3. Predictive Determinants of Sleep and Distress

At T0, prevalence of parental sleep problems (score >1SD above the reference’s mean) was 50%. At T3, this percentage was 33%, and 48% of these parents simultaneously reported clinical distress levels. Table 3 shows the final multinomial mixed-model analysis. Over time—compared to parents without clinical sleep problems and with low levels of distress—parents who reported both were more likely to perceive a lack of social support, experience parenting problems, report a chronic illness, report pain for their child, have a child with MR/HR-ALL, and be closer to diagnosis. The same risk factors (except for self-reported chronic illness) were found for parents who reported only distress, whereas the only identified risk factor in parents with sleep problems but without distress was presence of parenting problems.

### 3.4. Relationships between Sleep, Distress, and QoL

Figure 3 shows the average MCS z-scores over time, across sleep/distress categories. Parents with either sleep problems or clinical distress had a significantly lower z-score over time than parents without sleep problems or distress (mean difference −0.63 [−0.93; −0.34] and −0.90 [−1.12; −0.69], respectively—corresponding with SD-scores of −0.2 [−0.5; 0.0] and −0.5 [−0.7; −0.3]. Parents with both sleep problems and clinical distress had a cumulatively lower z-score over time (mean difference −1.74 [−1.96; −1.51], SD-score −1.3 [−1.5; −1.2], which was also significantly lower than the other two categories (parents with only sleep problems or only distress). No statistically significant relationship was found between sleep/distress categories and PCS z-score over time.

## 4. Discussion

### 4.1. Main Findings

This is the first study that aimed to longitudinally assess the course and interrelationships of sleep problems, distress, and QoL in parents of children with ALL, from diagnosis up to three years later. Additionally, this study assessed predictive determinants of experiencing sleep problems and/or distress over time. We found that distress and mental QoL gradually improved to normal levels from baseline (T0, post-induction) up to three years after diagnosis (T3). Sleep problems improved significantly from T0 to T3, but were still elevated at T3. Of the parents with sleep problems at T3, about half also reported clinical distress. Risk factors for reporting both sleep problems and distress over time were perceived lack of social support, experiencing parenting problems, self-reported parental chronic illness, parent-reported child’s pain, shorter time since diagnosis, and higher ALL risk group. Presence of sleep problems and clinical distress had a cumulative, adverse effect on parents’ mental QoL over time.

This study provides important information on parental psychosocial functioning in pediatric oncology. Since parental and child functioning are closely related, it is for the benefit of the whole family to address parental well-being [2,29]. It is known that parents are at risk for QoL impairment, both throughout and after treatment [3,4,30]. Some determinants of QoL are neither easy, nor possible, to improve (e.g., family’s socioeconomic status, child medical variables). It is therefore essential to explore aspects of parental well-being that are potentially modifiable. We show here that sleep problems and clinical distress levels both (cumulatively) affect parental QoL. Previous research has also defined sleep problems and distress as important determinants for adverse QoL outcomes, and acknowledged their close relationship [3,4,10,31,32]. Yet, to date, little was known on the course of sleep problems in parents of children with cancer, and the interrelationships with distress and QoL over time.

We found that sleep problems are prevalent in parents of pediatric ALL-patients, and only decline in a portion of parents. At three years after diagnosis (one year after end of treatment for the far majority), the prevalence was still 33%—twice as high as in the general population. Previous research on parental sleep problems in pediatric oncology found similar or even higher prevalence (around 50% during outpatient treatment, up to 70% in parents of children hospitalized for stem cell transplant); yet these studies had mostly small sample sizes, and none of them had a longitudinal design [7,8,9,33,34,35].

It is important to address sleep problems, and identify possible precipitating, predisposing, and perpetuating factors. An important perpetuating factor that we explored in this study is parental distress. Although on average—similar to in other studies [5,6,36]—distress declined over time to normal levels in our study population, a large proportion of parents with sleep problems also reported clinical distress. Hence, in these parents, ongoing distress was likely a perpetuating factor. This is in line with a previous study by our group in a different population, which found that 3.5 years after their child’s diagnosis, 37% of the parents reported sleep problems, of which 75% in concurrence with distress [10].

The relationship between sleep and distress is probably cyclic: high stress exposure precipitates insomnia complaints [37], which can lead to increased distress through increased sympathetic activity with the release of catecholamines and enhanced cortisol secretion [38], which in turn affects sleep. Sleep reactivity refers to the degree in which an individual’s sleep is disrupted by stress, which can either be psychological or environmental (e.g., change in sleep timing or setting) [16]. Sleep problems of people with low sleep reactivity are often not stress-related and tend to be less severe, whereas people with high sleep reactivity are at risk for chronic or recurring insomnia complaints. This could explain why not all parents with sleep problems experienced clinical distress.

Since half of the parents with sleep problems did not report ongoing distress, there are likely other perpetuating factors as well. In the group of parents with impaired sleep without distress, the only risk factor that we identified was parenting problems. Parents who sleep poorly are more likely to engage in dysfunctional parenting, including difficulties with enforcing rules and setting boundaries (permissive parenting style) [39,40]. In a pediatric ALL population, this permissive parenting has been previously linked to, amongst others, poor child’s sleep [41]. Since parental and child sleep are closely related [8], the link between parenting problems and impaired parental sleep might be partially explained by poor child sleep. Other important factors are likely unhealthy sleep habits and dysfunctional cognitions about sleep, and circadian rhythm disturbances [12]. However, we did not assess these in this study.

Predisposing factors that make parents more vulnerable to sleep problems and distress—besides sleep reactivity—are presence of chronic illness, little social support, parenting problems, and low socioeconomic status (SES). Low SES has been identified as risk factor in previous research [13], but did not emerge as a risk factor in our study. However, we only assessed family’s educational level as derivative of SES, and had an overrepresentation of highly educated families in our sample. Additionally, it would have been interesting to include the financial impact of pediatric ALL on parents. Previous research showed that the cancer-related financial burden for families can be severe, which is associated with increased parental distress and quality of life impairment [42,43,44].

The strongest predictors of parental sleep and distress are psychosocial factors, such as insufficient social support and parenting problems. These will not only be important for parents of children with leukemia, but for parents of children with all types of cancer diagnoses. Problems that were assessed included general parenting difficulties (e.g., child’s emotions, behavior, and independence), as well as specific difficulties for parents of chronically ill children (e.g., discussing the illness and consequences with the child, and administering medication). Parents could benefit from interventions that address these particular problems and empower them by teaching coping skills.

Finally, we found that medical factors are predictive of sleep problems and distress, e.g., shorter time since diagnosis, child’s symptoms (pain), and MR/HR-ALL as opposed to SR-ALL. Previous research has also shown that medical factors, such as risk group stratification—which reflects treatment intensity—may contribute to higher distress [45,46]. Particularly dexamethasone treatment in children with MR-ALL could influence parental psychosocial functioning [46,47,48]. Adequate supportive care and symptom management are of utmost importance.

### 4.2. Clinical Implications

This study has several clinical implications. First, it is important to screen for and monitor sleep problems in parents of children with cancer, and provide timely interventions. Sleep problems tend to be chronic and underdiagnosed, although evidence-based interventions exist, which could also benefit parental well-being [3,4]. During the first, intensive phases of treatment, intervention options might be limited to education on importance of sleep, and sleep hygiene advice. In later stages, if parents report ongoing sleep problems, first-line treatment of insomnia is cognitive behavioral therapy (CBT-i) [49]. This has been proven effective in a variety of populations, and consists of several key elements (e.g., altering dysfunctional sleep cognition, sleep restriction in order to increase sleep pressure, relaxation techniques) [15]. Since some parts of CBT-i include relaxation, it may simultaneously have positive effects on distress. However, sleep interventions have not yet been evaluated in parents of children with cancer, who might have specific needs with regard to stress and trauma management.

A recent intervention that has shown to be effective in providing psychosocial support to whole families is the FAMily-Oriented Support (FAMOS) intervention [50]. Other feasible interventions for parents are, for example, the Surviving Cancer Competently Intervention Program (SCIPP) [51] and Cascade [52], which provide parents with coping strategies to reduce their distress.

Second, it is important to assess the psychosocial risk profile of parents at an early stage. The above-mentioned predisposing factors, i.e., socioeconomic and psychosocial resources such as social support, can all be explored with the Psychosocial Assessment Tool (PAT) [53]. A previous study showed that this risk profile is indicative of the levels of distress that parents will experience during the treatment trajectory [54]. Findings of our study suggest that the PAT will also provide an indication of the risk for sleep problems, considering the close relationship with distress and their common predictive factors. Additionally, as stressed by the Standards of Psychosocial Care [2], this study highlights the need to systematically monitor parents’ mental health needs throughout and after treatment, and intervene timely.

### 4.3. Limitations

This study has some limitations. First, we did not have age- and sex-specific reference values of the MOS Sleep Scale, while these variables are known to influence sleep [55]. Second, parents in our sample were mainly highly educated and there was barely any variation in ethnicity. Similarly, only one family of a child with HR-ALL participated. These three factors might indicate some participation bias, and could have led to an underestimation of parental difficulties. Additionally, parent respondents were mostly mothers, which makes it harder to generalize our findings to fathers. Finally, with regards to sleep, we did not have any information on pre-existent sleep problems or prior episodes of sleep problems, neither on sleep habits or cognitions. This type of data may inform interventions, and should therefore be addressed in future research.

## 5. Conclusions

Parental well-being improves over time, yet sleep problems persist in a significant subset of parents. In psychosocial screening, special attention should be given to sleep in combination with ongoing distress, since this is a significant risk factor for QoL impairment. Additionally, family’s psychosocial risk profile should be assessed at an early stage, since this is indicative of future parental functioning.

## Figures and Tables

**Figure 1 cancers-14-02779-f001:**
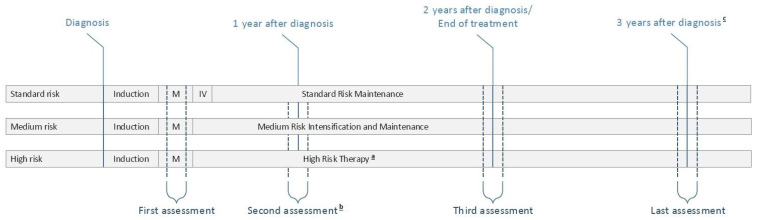
Schematic overview of DCOG ALL11 treatment protocol and SLAAP-study assessments. ^a^ High-risk therapy included an allo-SCT for the majority of patients. ^b^ Second assessment included an extra assessment for parents of MR-patients, during a week with dexamethasone (not taken into account in the longitudinal analyses). ^c^ Last assessment was one year after end of treatment for most patients, and around end of treatment for MR-patients with an IKZF1 deletion and a third year of treatment.

**Figure 2 cancers-14-02779-f002:**
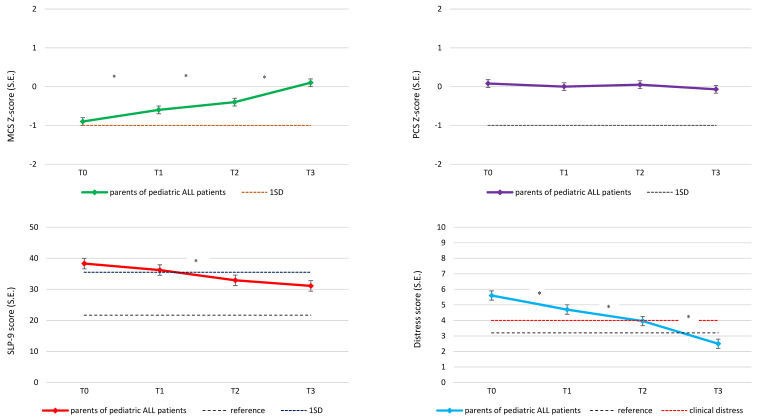
Linear mixed-models analysis: longitudinal course of parental sleep, distress, and quality of life (analyses are adjusted for parent’s sex and intercepts are displayed for mothers); * indicates a significant change (*p* < 0.05) between the two given time points. SLP-9: 9-item sleep problems index; PCS: Physical Component Summary; MCS: Mental Component Summary.

**Figure 3 cancers-14-02779-f003:**
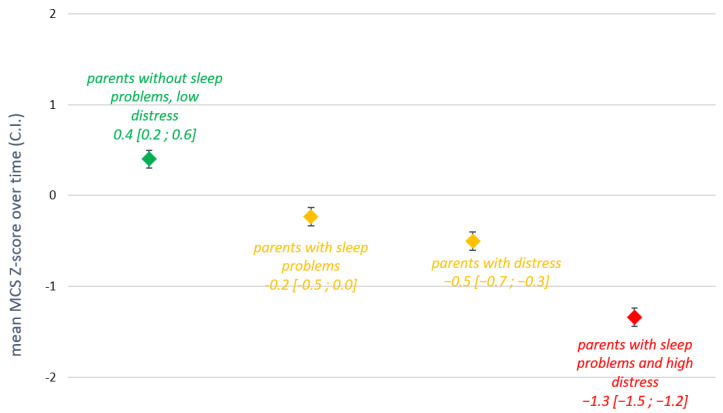
Linear mixed-models analysis: mean Mental Component Summary (MCS) z-score over time for parents with or without sleep problems and/or distress, corrected for parent’ sex.

**Table 1 cancers-14-02779-t001:** Patients’ baseline characteristics (*N* = 139).

**Sex**	***N* (%)**
Male	83 (59.7)
Female	56 (40.3)
**Risk group stratification**	***N* (%)**
Standard risk	32 (23.2)
Medium risk	105 (76.1)
*of which with confirmed IKZF1 deletion* *and third year of treatment*	*8 (7.6)*
High risk	1 (0.7)
Unknown (deceased before risk group stratification)	1 (0.7)
**Age at diagnosis**	**Median (interquartile range)**
*in years*	4.8 (3.1–8.7)
**Chronic illness (other than ALL)**	***N* (%)**
Yes	8 (5.8)
No	124 (89.2)
Unknown	7 (5.0)
**Family’s educational level**	***N* (%)**
Low	5 (3.6)
Middle	39 (28.1)
High	88 (63.3)
Unknown	7 (5.0)

**Table 2 cancers-14-02779-t002:** Parents’ characteristics—cross-sectional (per time point).

	T0 *N* = 120	T1 *N* = 112	T2 *N* = 101	T3 *N* = 92
**Parent’s sex**	*N* (%)			
male	26 (21.7)	19 (17.0)	18 (17.8)	16 (17.4)
female	94 (78.3)	92 (82.1)	82 (81.2)	75 (81.5)
unknown	0 (0.0)	1 (0.9)	1 (1.0)	1 (1.1)
**Parent’s age**	Mean (SD)			
in years	38.6 (6.4)	39.0 (6.1)	39.9 (6.4)	40.9 (6.0)
**Chronic illness**	*N* (%)			
yes	14 (11.7)	12 (10.7)	7 (6.9)	5 (5.4)
no	106 (88.3)	100 (89.3)	93 (92.1)	87 (94.6)
**Time since child’s ALL diagnosis**	Mean (SD)			
in months	4.7 (1.3)	13.5 (1.3)	24.2 (1.7)	36.7 (1.8)
**Parent-reported pain of the child**	*N* (%)			
clinically relevant pain score (≥4)	54 (45.0)	36 (32.1)	33 (32.7)	20 (21.7)
no clinically relevant pain score (<4)	65 (54.2)	72 (64.3)	68 (67.3)	68 (73.9)
unknown	1 (0.8)	4 (3.6)	0 (0.0)	4 (4.3)
**SLP-9**				
mean score (SD)	36.4 (16.7)	34.3 (18.3)	30.9 (17.1)	28.4 (16.4)
% clinically relevant sleep problems	51.7	40.2	40.0	32.6
**Distress**				
mean thermometer score (SD)	5.4 (2.8)	4.5 (2.4)	3.9 (2.8)	2.4 (2.5)
% clinical distress	72.0	66.7	51.1	26.8
**Quality of life**				
mean MCS z-score (SD)	−0.9 (1.2)	−0.5 (1.1)	−0.4 (1.0)	0.1 (1.0)
% clinically impaired	48.3	36.1	29.3	11.5
mean PCS z-score (SD)	0.1 (1.2)	0.0 (1.0)	0.2 (0.9)	0.1 (0.8)
% clinically impaired	14.7	13.0	10.1	9.2
**Parenting problems**	*N* (%)			
yes	56 (46.7)	49 (43.8)	42 (41.6)	23 (25.0)
no	63 (52.5)	63 (56.3)	58 (57.4)	67 (72.8)
unknown	1 (0.8)	0 (0.0)	1 (1.0)	2 (2.2)
**Social support**	*N* (%)			
sufficient	106 (88.3)	94 (83.9)	82 (81.2)	82 (89.1)
insufficient	14 (11.7)	17 (15.2)	18 (17.8)	10 (10.9)
unknown	0 (0.0)	1 (0.9)	1 (1.0)	0 (0.0)
**Wish for referral**	*N* (%)			
yes/maybe	51 (42.5)	43 (38.4)	27 (26.7)	20 (21.7)
no	67 (55.8)	69 (61.6)	73 (72.3)	70 (76.1)
unknown	2 (1.7)	0 (0.0)	1 (1.0)	2 (2.2)

**Table 3 cancers-14-02779-t003:** Multinomial mixed-model analysis: predictors of sleep problems and distress over time, per group—corrected for parent’s sex.

	Sleep Problems, Low Distress ^a^	High Distress, No Sleep Problems ^a^	Sleep Problems and High Distress ^a^
**Parent variables**	**OR [95% C.I.]**	**OR [95% C.I.]**	**OR [95% C.I.]**
Chronic illness	2.2 [0.4; 12.4]	1.2 [0.3; 5.5]	3.7 [0.9; 15.9] *
**Child variables**			
TSD (per one year increase)	0.9 [0.6; 1.4]	0.5 [0.4; 0.7] ****	0.5 [0.3; 0.7] ****
Medium or high risk group	1.0 [0.4; 2.6]	2.1 [0.9; 4.7] *	3.0 [1.2; 7.7] **
Clinically relevant pain (parent-rated)	0.5 [0.2; 1.6]	2.7 [1.3; 5.5] ***	4.3 [2.0; 9.5] ****
**Psychosocial variables**			
Parenting problems	2.3 [0.9; 5.7] *	3.8 [1.9; 7.7] ****	4.5 [2.1; 9.5] ****
Insufficient social support	1.8 [0.3; 11.0]	4.9 [1.2; 19.8] **	15.2 [3.8; 61.2 ] ****

^a^ compared to parents without sleep problems or distress (reference); * *p* < 0.10; ** *p* < 0.05; *** *p* < 0.01; **** *p* < 0.001.

## Data Availability

The data of this study are available from the corresponding author, upon reasonable request.

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
