# Peer review of "Parental Sleep, Distress, and Quality of Life in Childhood Acute Lymphoblastic Leukemia: A Longitudinal Report from Diagnosis up to Three Years Later"

_cancers, 2022, doi:10.3390/cancers14112779_

Round 1

Reviewer 1 Report

Review on Parental sleep, distress and quality of life in childhood acute lymphoblastic leukemia: a longitudinal report from diagnosis up to three years later

The authors present a longitudinal-sectional cohort study using population-based data. The study is designed to investigate trends of sleep problems, distress, and quality of life in parents of children with acute lymphoblastic leukemia (ALL) and to identify associated factors. Data of 139 parents of children with ALLs were included. The authors found that parental functioning improves over time, although 1/3 of the parents still report sleep problems three years after their child’s diagnosis. Furthermore, the authors identified risk factors like social support, parenting problems, report on chronic illness for themselves or pain for their child, having a child with higher risk ALL, and are closer to the diagnosis. This is an interesting paper with relevance to researchers and clinicians working in this field. The article is well presented and a good fit for the journal. I have, however, some comments to do:

Minor comments and revisions

  1. The research questions and aims in the introduction lack clarity and therefore should be guided by approaches like PICO criteria.
  2. In the methods section more information on the inclusion criteria should be given.
  3. In the methods section the authors refer to a description of the study population in the form of “as describe in detail elsewhere (p.1, line 85). For clarity, a corresponding reference should be given. The same applies to the description of the questionnaires (p.3)
  4. In the methods section and the results section, the p-value should set in italics.
  5. In the results section, it is not clear why the authors decided to use the rather unconventional approach to use a p-value of <.10 considered to be significant. Please explain.

Reviewer 2 Report

Thanks for your invitation to review this manuscript. The authors designed a cohort study to examine the change in sleep quality in patients of children with ALL. 

Pleaes find my comments below:

1. Please briefly describe the study population and questionnaires. They are critical for readers to understand how subjects were selected. 
2. I would not recommend to using backward selection in the multiple models, as it results in unstable models with subgroup analyses. 
3. The authors may like to describe the difference between medium-risk and low-risk maintenance therapies as most readers may not know the difference. 
4, I am wondering if the authors looked at any interaction between subjects' characteristics such as gender, support, and parenting problems, and how these interactions impact the sleep quality. It would also be helpful to understand what parents would want referral as this is clinically relevant. 
5. It is a bit odd to see the authors chose both sleep quality and distress as outcomes. They may like to focus on sleep quality itself and examine the impact of distress on this model. It would help readers to better understand the role of distress on patients' sleep quality.
6. While the authors' study did not examine it, they may like to discuss the financial impact on parents' distress level and sleep quality. 
